TOPICAL REVIEW

# Involvement of the extracellular matrix and integrin signalling proteins in skeletal muscle glucose uptake

Fulvia Draicchio[1] (iD), Volker Behrends[1], Neale A. Tillin[1] (iD), Nicholas M. Hurren[1], Lykke Sylow[2] (iD) and Richard Mackenzie[1] (iD)

[1] *School of Life and Health Sciences, Whitelands College, University of Roehampton, London, UK*
[2] *Molecular Metabolism in Cancer & Ageing Research Group, Department of Biomedical Sciences, University of Copenhagen, Copenhagen, Denmark*

Handling Editors: Ian Forsythe & Javier Gonzalez

The peer review history is available in the Supporting information section of this article (https://doi.org/10.1113/JP283039#support-information-section).

**Abstract** Whole-body euglycaemia is partly maintained by two cellular processes that encourage glucose uptake in skeletal muscle, the insulin- and contraction-stimulated pathways, with research suggesting convergence between these two processes. The normal structural integrity of the

**Fulvia Draicchio** had a background in biotechnology and genetics before moving into the research theme of human metabolism and metabolic disorders of type 2 diabetes and chronic kidney disease in the Laboratory of Dr Richard Mackenzie in collaboration with Dr Nick Burd. Her research suggests that disruption to the transmembrane protein linkage between the cytoskeleton and the extracellular matrix in skeletal muscle may contribute to reduced amino acid metabolism and insulin resistance in haemodialysis. **Richard W. A. Mackenzie** is a Reader in Insulin Resistance and Metabolism at the University of Roehampton (London). With over 10 years' experience in investigating the mechanisms that govern contraction- and insulin-stimulated glucose transport in skeletal muscle, he has a particular interest in the role of inositol hexakisphosphate (IP6) kinase 1 (IP6K1) in the inhibition of Akt/protein kinase B and reduced insulin signalling in skeletal muscle. His research focus requires molecular and whole-body approaches to investigate the disease processes in human metabolism and type 2 diabetes.

skeletal muscle requires an intact actin cytoskeleton as well as integrin-associated proteins, and thus those structures are likely fundamental for effective glucose uptake in skeletal muscle. In contrast, excessive extracellular matrix (ECM) remodelling and integrin expression in skeletal muscle may contribute to insulin resistance owing to an increased physical barrier causing reduced nutrient and hormonal flux. This review explores the role of the ECM and the actin cytoskeleton in insulin- and contraction-mediated glucose uptake in skeletal muscle. This is a clinically important area of research given that defects in the structural integrity of the ECM and integrin-associated proteins may contribute to loss of muscle function and decreased glucose uptake in type 2 diabetes.

(Received 28 February 2022; accepted after revision 3 August 2022; first published online 2 September 2022)
**Corresponding author** R. Mackenzie: School of Life and Health Sciences, Whitelands College, Holybourne Avenue, London SW15 4DJ, UK. Email: richard.mackenzie@roehampton.ac.uk

**Abstract figure legend** Nutrient and hormonal flux impeded by the accumulation of excessive extracellular matrix (ECM) proteins, causing a physical barrier for glucose uptake in skeletal muscle. Akt, protein kinase B; FAK, focal adhesion kinase; ILK, integrin-linked kinase; IRS1, insulin receptor substrate 1; NCK2, non-catalytic region of tyrosine kinase adaptor protein 2; PI3K, phosphatidylinositol 3-kinase; PINCH, particularly interesting new cysteine–histidine-rich protein; Rac1, Ras-related C3 botulinum toxin substrate 1.

## Introduction

Glucose homeostasis involves a complex interplay and cross-talk between pancreatic $\beta$-cells and insulin-sensitive tissues such as skeletal muscle, adipose tissue and the liver. This cross-talk is defective in type 2 diabetes (T2D) due to a combination of failing $\beta$-cell function, reduced insulin sensitivity and elevated endogenous glucose production (Stumvoll et al., 2005). Reduced insulin sensitivity is considered the starting point for the development of T2D, with $\beta$-cell dysfunction presenting at the latter stages of the progression from pre-diabetes to overt T2D (Stumvoll et al., 2005). Human studies show that skeletal muscle is the primary site for insulin-mediated glucose disposal (Baron et al., 1991a; Bouché et al., 2004; DeFronzo et al., 1985). Thus, a substantial amount of research has been directed towards understanding the molecular defects associated with insulin signalling in skeletal muscle to develop novel treatment strategies for conditions such as T2D. Yet, to date there are no drugs available that target skeletal muscle insulin resistance and therefore it is important to clarify the underlying molecular mechanisms. As canonical insulin signalling has not presented optimal targets, lesser studied pathways, such as the actin cytoskeleton, extracellular matrix (ECM) and focal adhesions, might present new opportunities.

The causes of insulin resistance and reduced insulin-stimulated glucose uptake in skeletal muscle are complex and likely involve disruptions to (1) glucose delivery, (2) transport across the sarcolemma and (3) glucose metabolism with defects in the intracellular protein networks contributing to failure of the second and third of these processes (Karlsson et al., 2005; Mackenzie & Elliott, 2014; Zick, 2001). Importantly, and related to the three points above, mounting evidence in rodents and humans supports the notion that the actin cytoskeleton and ECM may play an important role in muscle glucose uptake and GLUT4 transport (Berria et al., 2006; Dixon et al., 2013; Inoue et al., 2013; Kang et al., 2011, 2013, 2014; Richardson et al., 2005; Wada et al., 2013; Williams et al., 2015). Moreover, the major ECM surface receptors integrins, and their downstream effectors, are emerging as potential players in skeletal muscle insulin action and glucose uptake (Williams et al., 2015). Therefore, understanding the role that integrin and its associated structural proteins plays in insulin action and glucose transport in skeletal muscle may be fundamental to identify new therapeutic targets for the treatment of insulin resistance and T2D. This review will outline and discuss the emerging evidence for the ECM and its associated integrin proteins playing a critical role in skeletal muscle glucose uptake.

## The canonical pathways responsible for facilitating glucose uptake in skeletal muscle

Skeletal muscle provides a large disposal site for both the oxidation and the storage of glucose, making it a key tissue in the management of diseases such as T2D (DeFronzo et al., 1982). The plasma membrane of the skeletal muscle fibre is unique in that it consists of the sarcolemma and the transverse tubules (T-tubules). In the absence of insulin and at rest, muscle glucose transport likely occurs via glucose transporter (GLUT) 1 (Marette, Richardson et al., 1992; Rudich et al., 2003), localised constitutively at the sarcolemma.

In skeletal muscle, two key stimuli facilitate glucose uptake: insulin and contraction (Holloszy, 2003; Kjøbsted et al., 2019; Lundell & Krook, 2013; Richter et al., 1985; Wallberg-Henriksson & Holloszy, 1985). Yet rapidly evolving research reports significant overlap between contraction- and insulin-stimulated glucose transport, suggesting convergence between these pathways (Cortright & Dohm, 1997; Ihlemann et al., 2000; Kim et al., 2021; Richter et al., 2021; Wojtaszewski et al., 1996). In addition, exercise is known to stimulate over 1000 phosphorylation sites in human skeletal muscle, suggesting that there are a great number of regulators yet to be identified (Sylow et al., 2016). The T-tubules likely represent a major site of glucose transport in skeletal muscle, as the majority of GLUT4 mobilised in response to insulin inserts into the T-tubule membranes and only a smaller fraction into the sarcolemma in human and rodent muscle (Henríquez-Olguin et al., 2019; Knudsen et al., 2020; Lauritzen, 2013; Marette, Burdett et al., 1992). Upon stimulation, GLUT4s translocate from GLUT storage vesicles in the cytoplasm to the cell surface to facilitate glucose uptake across the sarcolemma (Sylow et al., 2021). The processes involved in GLUT4 trafficking have received a great deal of attention and have been reviewed elsewhere (McConell et al., 2020; Merz et al., 2022; Richter, 2021; Richter & Hargreaves, 2013; Sylow et al., 2017, 2021; Tunduguru & Thurmond, 2017), and therefore they will only be briefly outlined within this review.

Upon binding to its receptor, insulin initiates a cascade of events at the muscle cell membrane that results in the phosphorylation of insulin receptor substrate 1 (IRS-1) and the subsequent activation of phosphatidylinositol 3-kinase (PI3K) (Sylow et al., 2021). At this intersection, PI3K offers an important bifurcation point leading to two parallel signalling cascades required for GLUT4 translocation, one requiring the serine–threonine kinase Akt1/2 (Brozinick & Birnmbaum, 1998; Lund et al., 1998; Wang et al., 1999) and the other, the Rho family GTPase Rac1 (JeBailey et al., 2007; Sylow et al., 2013, 2014, 2021; Ueda et al., 2008). Downstream of Akt, the phosphorylation of TBC1 domain family member 4 (TBC1D4 or AS160) leads to TBC1D4 inhibition, and the subsequent release of the intracellular GLUT storage vesicles. Together with Rac1-mediated cytoskeletal reorganisation, these signalling pathways promote GLUT4 translocation and increased glucose uptake in response to insulin. (For a recent review see Sylow et al., 2021.)

Contraction of skeletal muscle also provides a potent mechanism for GLUT4 translocation and glucose uptake. This process is independent of insulin and seems to require, in part, AMP-activated protein kinase (AMPK) and Rac1 (Hardie & Lin, 2017; Ito et al., 2005; Kjøbsted et al., 2017, 2019; Sylow et al., 2016; Yue et al., 2020).

With cellular changes in energy status, and increased excitation–contraction coupling, both elevated $Ca^{2+}$ and AMP:ATP ratios are known activators of AMPK (Hawley et al., 1996; Hayashi et al., 1998; Jensen et al., 2007, 2014; Rose & Richter, 2005) resulting in TBC1D4Ser704 phosphorylation via the AMPK $\gamma$3-containing complexes (Blair et al., 2009; Chen et al., 2008, 2011; Cheung et al., 2000; Eickelschulte et al., 2021; Hardie & Carling, 1997), allowing for GLUT4 translocation to the cell membrane (Kjøbsted et al., 2019). (For recent reviews see Richter et al., 2021; Sylow et al., 2017.) Emerging evidence suggests substantial cross-talk between these canonical pathways and the ECM in facilitating glucose transport in skeletal muscle, which will be outlined in the following sections.

## Extracellular matrix and transmembrane protein structure

The ECM network on the plasma membrane is linked to the intracellular cortical actin cytoskeleton in the cytoplasm through a nexus of proteins. These proteins include integrins, talin, vinculin, focal adhesion kinase (FAK), Arp 2/3 and Rac1 (Csapo et al., 2020; Delon & Brown, 2007; DeMali et al., 2002; Hsiao et al., 2015; Vicente-Manzanares et al., 2009), which are located downstream of the ECM and interact with the transmembrane integrins (Fig. 1). The ECM is a dynamic structure that consists of a network of proteins that modulate biological processes including insulin-stimulated glucose transport, cell migration, differentiation, development and repair (Andez & Amenta, 1995; Hynes, 2009; Schuppan, 1990; Williams et al., 2015). Moreover, the ECM is fundamental for cell-to-cell interaction as well as function and maintenance of all tissue (Mayer, 2003; Williams et al., 2015). The fabrication of the ECM consists of over 300 proteins, with collagens, proteoglycans, laminin, integrins, elastin and cell-binding glycoproteins forming the major elements of the ECM, each with distinct physical and biochemical properties (Hynes & Naba, 2011; Hall & Sanes, 1993).

Collagen is the most abundant ECM structural component and is essential for cell adhesion, migration, differentiation, wound healing and morphogenesis (Aumailley & Gayraud, 1998; Kang et al., 2011; Zutter & McCall-Culbreath, 2008). Moreover, collagens are required for tissue support and structural integrity (Aumailley & Gayraud, 1998; Kang et al., 2011). Collagen I, III and IV are the most abundantly expressed isoforms in mammalian skeletal muscle, with the latter found mostly in the basement membrane (Kang et al., 2011; Yurchenco & Patton, 2009).

Proteoglycans are protein structures that are heavily glycosylated, formed of a core protein with one or more glycosaminoglycan (GAG) side chains attached; they are

discussed elsewhere (Yue, 2014). Importantly for skeletal muscle, proteoglycans form a network with growth factors and growth factor receptors, including the insulin receptor (Morcavallo et al., 2014; Ohta et al., 2018; Ussar et al., 2012). They are associated in cell signalling and biological processes, including angiogenesis (Yue, 2014), and may have a role in promoting glucose transport and diffusion.

The dystrophin–glycoprotein complex (DGC) is a large multicomponent membrane-spanning complex that binds extracellular proteins with intracellular non-contractile proteins (Barresi & Campbell, 2006; Langenbach & Rando, 2002), thus providing a physical structure between the basement membrane and the subsarcolemmal cytoskeleton (Fig. 1). The central component of the DGC is the dystroglycan protein, consisting of two glycoprotein subunits, $\alpha$ and $\beta$ (Holt et al., 2000; Ibraghimov-Beskrovnaya et al., 1992; Langenbach & Rando, 2002), with each playing a distinct role in muscle structure and stability, as already described elsewhere (Ervasti & Campbell, 1991; Ibraghimov-Beskrovnaya et al., 1992). $\alpha$-Dystroglycan, binds to the ECM via various laminin isoforms (laminin 1, 2 and 4) on one side, and to the cortical actin cytoskeleton on the other side, through $\beta$-dystroglycan (Holt et al., 2000; Ibraghimov-Beskrovnaya et al., 1992; Langenbach & Rando, 2002). $\beta$-Dystroglycan is a transmembrane protein bound to $\gamma$-actin via the large non-contractile protein dystrophin, located intracellularly (Ervasti & Campbell, 1991; Langenbach & Rando, 2002; Michele & Campbell, 2003; Rybakova et al., 2000). Interestingly, the DGC plays a protective role in preventing actin depolymerisation (Ervasti, 2013). This is an important point given that actin depolymerisation and optimal actin reorganisation are needed for insulin- (Brozinick et al., 2004; JeBailey et al., 2004; JeBailey et al., 2007; Sylow et al., 2013, 2014) and contraction- (Sylow et al., 2012) mediated glucose uptake in skeletal muscle.

The ECM communicates using transmembrane cell surface receptors known as integrins (Mayer, 2003). Laminin acts as a ligand for the integrin receptor (Grounds et al., 2005), positioned at costameres and aligned with the Z-disks of the myofibrils (Csapo et al., 2020). In forming a link between the ECM and the cytoplasm, integrins transduce signals across the plasma membrane and activates intracellular signalling, including several downstream kinases that appear to be implicated in nutrient uptake and glucose metabolism (Williams et al., 2015). Integrins are heterodimeric, transmembrane glycoproteins with an $\alpha$ and a $\beta$ chain non-covalently associated (Mayer, 2003; Rahimov & Kunkel, 2013). Thus far, 18 $\alpha$ and eight $\beta$ chains have been identified, which combine to form at least 24 different dimers (Mayer, 2003; Van der Flier & Sonnenberg, 2001). The diversity of these proteins is increased by the expression of splice variants that generates sub-chains. Seven $\alpha$ subunits are expressed in skeletal muscle, $\alpha$1, $\alpha$3, $\alpha$4, $\alpha$5, $\alpha$6, $\alpha$7 and $\alpha$V, all of which are linked to the $\beta$1 integrin subunit (Williams et al., 2015). Integrins are known to provide a bidirectional linkage between the ECM and the cytoskeleton, transferring external stimuli to regulate cellular processes while also allowing intracellular signalling proteins to contribute to external adhesion (Boppart & Mahmassani, 2019; Csapo et al., 2020). Integrins lack endogenous enzymatic activity, so they are believed to form focal adhesions, composed of complex groups of downstream signalling molecules and proteins to provide a link to actin and microtubule cytoskeletons (Clemente et al., 2012; Hynes, 2002; Schober et al., 2007). Integrins form a structural linkage with the actin cytoskeleton including the actin-binding protein talin, which links to the $\beta$ subunit of integrin, contributing to force development within the actin cytoskeleton (Gheyara et al., 2007).

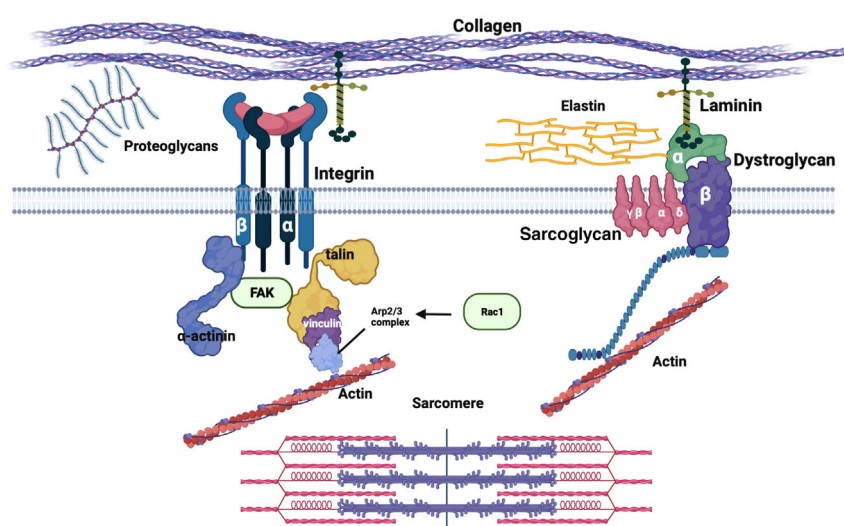

**Figure 1. Key components of the extracellular matrix (ECM) of skeletal muscle, demonstrating the linkage between the ECM-associated proteins structures and the contractile filaments of muscle**
Abbreviations: Arp2/3complex, actin-related protein 2/3 complex; FAK, focal adhesion kinase.

In the context of this review, it seems prudent to highlight the positioning of Rac1 within ECM–actin cytoskeleton arrangements given its central role in actin cytoskeleton remodelling and GLUT4 translocation. In response to stimuli that generate membrane expansion, the actin-related protein 2/3 (Arp 2/3) complex binds to vinculin, an interaction regulated by phosphatidylinositol-4,5-bisphosphate and Rac1 (Fig. 1) (DeMali et al., 2002). The activation of the Arp 2/3 complex by Rac1 and the Scar/WAVE proteins (Machesky & Insall, 1998; Miki et al., 1998) results in the complex binding to actin (DeMali et al., 2002). Importantly, the involvement of Rac1 in the linkage of integrin–talin–vinculin offers a clear connection between the structural complements of skeletal muscle and the cellular processes involved in glucose transport activity.

## Integrin-associated proteins and the actin cytoskeleton: implications for glucose uptake

Although the role of integrins and ECM-associated proteins in ECM remodelling and muscle cell integrity has been extensively characterised (Draicchio et al., 2020; Legate & Fässler, 2009; Wickström et al., 2010), little is known about the specific functions of the intracellular protein complexes within the context of muscle structural stability and the regulation of insulin-stimulated cytoskeleton remodelling and glucose transport (Draicchio et al., 2020; Gheyara et al., 2007). It has been challenging to study the physiological role of integrins because it is difficult to recreate the complex ECM and cell–cell junction environment *in vitro*, and thus good model systems are still needed. Moreover, several of the key proteins involved in muscle structural stability seem to not have a catalytic activity (e.g. integrin $\beta$1 subunit, ILK, NCK).

Skeletal muscle actin cytoskeleton consists of $\beta$- and $\gamma$-actin. The actin cytoskeleton undergoes substantial reorganisation in response to insulin stimulation (Dugina et al., 2009; Kee et al., 2009), in a Rac1-dependent manner via cofilin and Arp (Chiu et al., 2010; JeBailey et al., 2007; Sylow et al., 2014), a process that mediates the very last steps of GLUT4 fusion with the plasma membrane (Brozinick et al., 2004; Brozinick et al., 2007). Accordingly, Rac1 is activated in response to insulin stimulation in skeletal muscle. Rac1 and the actin cytoskeleton likely act in parallel to Akt since individual inhibition of Rac1 or Akt partially decreased insulin-stimulated glucose transport by ∼40% and ∼60%, respectively. Yet, their simultaneous inhibition completely blocked insulin-stimulated glucose transport. Accordingly, the actin cytoskeleton depolymerising agent latrunculin B plus Akt inhibition blocked insulin-stimulated glucose uptake, while latrunculin B had no additive effect on Rac1 inhibition.

Emerging evidence suggests that integrin and its associated downstream targets may play a fundamental role in glucose metabolism in skeletal muscle (Bisht et al., 2008; Graae et al., 2019; Zong et al., 2009). This hypothesis is based on the following observations: (1) integrin-associated proteins and Rac1 seem to play a pivotal role in promoting actin remodelling that contributes to glucose uptake in skeletal muscle; and (2) integrin-associated proteins provide an essential linkage between the ECM, the sarcolemma and the actin cytoskeleton (Draicchio et al., 2020; Gheyara et al., 2007), thereby contributing to ECM structure stability, which may be necessary for normal nutrient transport in this tissue type (Draicchio et al., 2020; Gheyara et al., 2007).

Downstream integrin substrates include integrin-linked kinase (ILK), FAK (Draicchio et al., 2020; Hynes, 2002; Schober et al., 2007), Akt and Rac1 (Draicchio et al., 2020; Williams et al., 2015). The serine/threonine kinase ILK might be implicated in insulin-stimulated glucose transport. ILK complexes with particularly interesting new cysteine–histidine-rich protein (PINCH) and parvin forming the ILK–PINCH–parvin complex, which functions at the earliest steps of integrin signalling (Draicchio et al., 2020; Stanchi et al., 2009; Wu, 1999). This complex links integrins to the actin cytoskeleton through a number of insulin-sensitive downstream effectors, including parvin, $\alpha$-actinin, talin, Arp2/3 and the PI3K–Akt–Rac1 pathway (Fig. 2*A*) (Gheyara et al., 2007; Qian et al., 2005). ILK is ubiquitously expressed in mammalian tissues and has three distinct domains: (1) an N-terminal domain, (2) a pleckstrin homology-like domain, and (3) a C-terminal kinase-like domain (Stanchi et al., 2009; Wickström et al., 2010). ILK mediates its interactions with PINCH through the N-terminal domain, and parvin through the C-terminal domain (Legate et al., 2006; Stanchi et al., 2009; Wickström et al., 2010).

ILK interacts directly with the $\beta$-integrin subunits and recruits downstream targets implicated in the insulin-stimulated glucose uptake pathway, via IRS-1/Akt as well as the actin cytoskeleton remodelling via Rac1 (Draicchio et al., 2020; Gheyara et al., 2007). Specifically, ILK seems to activate 3-phosphoinositide-dependent protein kinase-1(PDK1) and glycogen synthase kinase 3 $\beta$ (GSK3$\beta$) through PINCH, thereby acting as an upstream regulator of Akt in skeletal and smooth muscle as well as other cell types (Qian et al., 2005; Tang et al., 2007; Williams et al., 2015; Wu & Dedhar, 2001). In addition, ILK recruits the downstream effectors $\alpha$-actinin and Rac1 through parvin (Draicchio et al., 2020; Williams et al., 2015) (Fig. 2*A*). This is likely implicated in promoting glucose uptake via cytoskeletal rearrangements because siRNA-mediated knockdown of ILK reduces Rac1

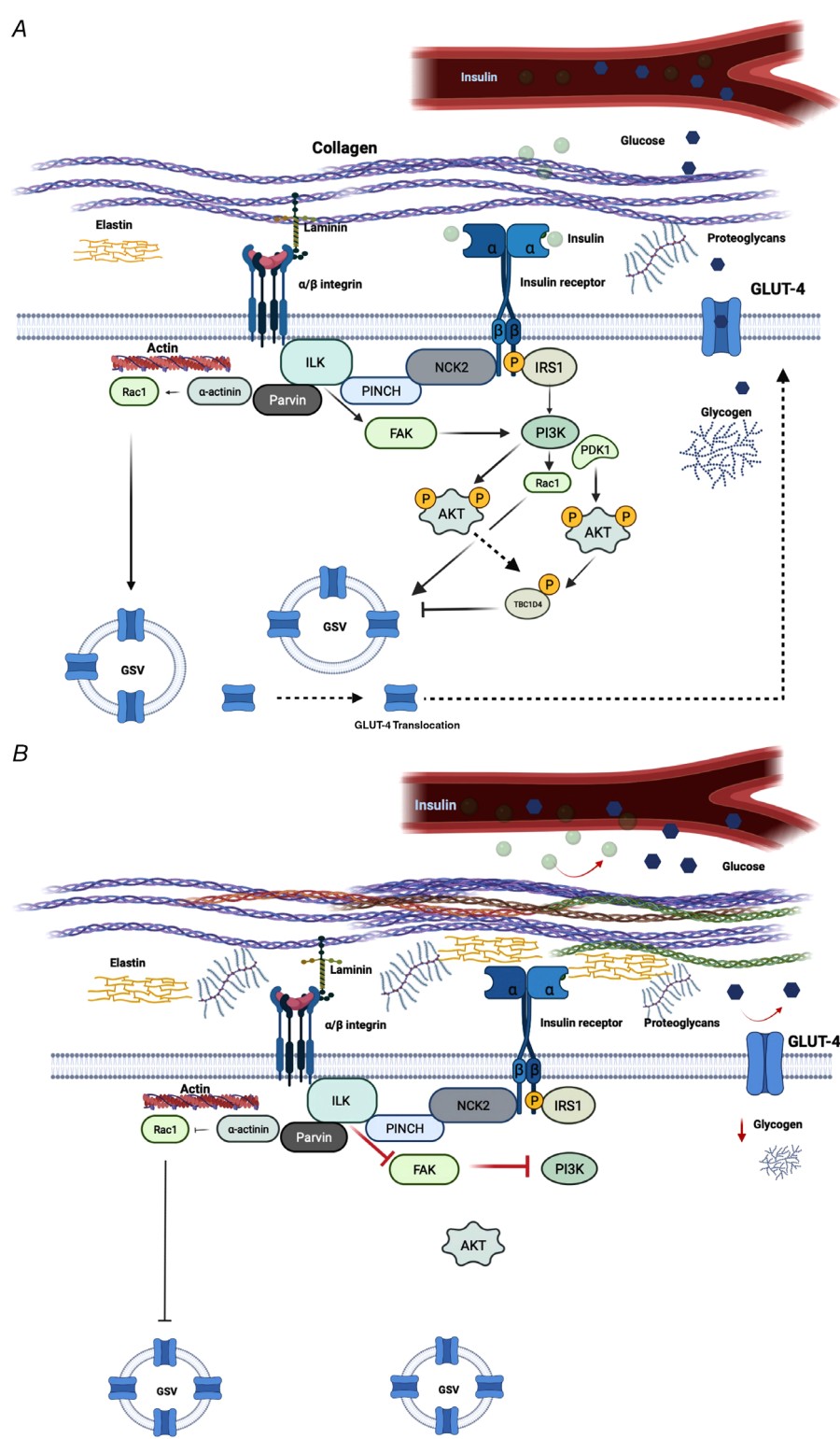

**Figure 2. Potential pathway linking integrins and their associated proteins in the regulation of glucose metabolism in muscle.**
*A*, Normal ECM synthesis leads to normal insulin and glucose delivery to the muscle. *B*, proposed disruptions to insulin signalling and glucose transport due to excessive ECM remodelling. Abbreviations: Akt, protein kinase B; GSV, GLUT4 storage vesicles; ILK, integrin-linked kinase; IRS1, insulin receptor substrate 1; PINCH, particularly interesting new cysteine–histidine-rich protein; Rac1, Ras-related C3 botulinum toxin substrate 1; TBC1D4, TBC1 domain family member 4.

activation and cytoskeleton reorganisation in epithelial cells (Filipenko et al., 2005).

FAK, together with ILK and the ILK–PINCH–parvin complex, appears to be another molecule that exerts the integrins' function with regard to the actin cytoskeleton (Draicchio et al., 2020; Williams et al., 2015). FAK is a non-receptor tyrosine kinase known to play an essential role in insulin signalling (Bisht et al., 2007; Huang et al., 2006; Parsons, 2003) via PI3K and its downstream substrates Akt (Bisht et al., 2008) and TBC1D4 (AS160) (Deshmukh et al., 2006; Geraghty et al., 2007; Rondas et al., 2012) (Fig. 2*A*). Once integrins are engaged and bind to the ECM, FAK is activated and interacts with the $\beta$1 integrin subunits and other signalling and cytoskeletal molecules at the focal adhesion sites (Chen et al., 2000; Schober et al., 2007). In addition, reducing FAK expression in L6 myotubes reduced glucose uptake, GLUT4 translocation, phosphorylated (p)-FAK$^{Tyr397}$, and actin fibre rearrangements under maximal insulin stimulation (100 nmol/l) (Huang et al., 2006). Under both control and maximal insulin treatment, glucose uptake and GLUT4 translocation were increased in C2C12 myocytes overexpressing FAK, implying that increased FAK protected C2FAK$^{wt/+}$ (FAK overexpressing) cells from hyperinsulinaemia-induced insulin resistance, whereas C2FAK$^{mut/+}$ (FAK tyrosine-397 mutated) cells developed insulin resistance. Furthermore, in the C2siRNAFAK$^{wt}$ (FAK siRNA cells) insulin-stimulated glucose uptake and GLUT4 translocation were significantly impaired under control and insulin treatment, supporting the notion that FAK elimination reduces glucose uptake and transport, likely as a consequence of impaired insulin signalling (Bisht et al., 2007), perhaps via a reduction in FAK-mediated PI3K–Akt activation and TBC1D4 inhibition (Bisht et al., 2008; Rondas et al., 2012).

## A potential role of extracellular matrix remodelling in diet-induced insulin resistance and reduced glucose uptake in muscle

Over the past decade, mounting evidence in humans (Berria et al., 2006) and rodents (Kang et al., 2011) suggests that increased ECM remodelling is associated with a number of pathological states, including insulin resistance in skeletal muscle. ECM proteins including fibronectin, proteoglycans and connective tissue growth factors increase severalfold in human (Berria et al., 2006; Richardson et al., 2005) as well as mouse skeletal muscles (Huber et al., 2007; Kang et al., 2011, 2013, 2014), adipose tissue (Inoue et al., 2013) and liver (Dixon et al., 2013; Wada et al., 2013), in response to diet-induced obesity (DIO). A high-fat diet (HFD) is associated with chronic systemic inflammation (Duan et al., 2018; Tan & Norhaizan, 2019), which increases ECM protein

synthesis as well as decreasing ECM protein degradation, resulting in increased deposition and remodelling of ECM (Ruiz-Ojeda et al., 2019) (Fig. 2*B*).

There are two working hypotheses that may explain the mechanisms underlying ECM-associated insulin resistance in DIO. The first is that the increased protein expression within the ECM creates a physical barrier preventing normal insulin action and glucose diffusion across the sarcolemma (Kang et al., 2013; Williams et al., 2015) (Fig. 2*B*). The second hypothesis suggests that muscle ECM may expand to impair vascular function and neovascular growth, given the close contact between the ECM and the endothelium (Williams et al., 2015). The first of these hypotheses suggests that collagen, fibronectin and proteoglycan proteins accumulate in the interstitial space increasing diffusion distance and impeding substrate and hormonal delivery (Berria et al., 2006; Williams et al., 2015). In support of this hypothesis, Kang et al. (2013) showed that hyaluronan (a major ECM component) in skeletal muscle was significantly increased in the insulin-resistant DIO mice when compared to chow-fed mice. Interestingly, the same authors also showed that treatment with long-acting pegylated human recombinant PH20 hyaluronidase (PEGPH20) caused a dose-dependent reduction in muscle hyaluronan content and improved skeletal muscle insulin resistance in DIO mice (Kang et al., 2013). These results show that whole-body depletion of an ECM polysaccharide improves muscle insulin sensitivity in obese mice, whereas ECM protein accumulation seems to exacerbate insulin resistance (Kang et al., 2013).

The second hypothesis that may explain the role the ECM plays in insulin resistance centres on the notion that nutrient delivery to the contracting muscle requires functional blood flow to ensure sufficient glucose (during exercise) and insulin (post-exercise) availability to facilitate glucose uptake and glycogen resynthesis, respectively. Thus, ECM vascular dysfunction and capillary rarefaction have been linked to insulin resistance and T2D (Jansson, 2007; Williams et al., 2015). In muscle, nutrient blood flow is enhanced by unperfused capillaries recruited by insulin (Bonner et al., 2013). Research shows that 40% of insulin-stimulated glucose uptake is attributed to augmented muscle perfusion, with this haemodynamic response known to be absent in insulin-resistant T2D patients (Baron et al. 1991b, 2000; Bonner et al., 2013; Ellmerer et al., 2006; Kim et al., 2008; Kubota et al., 2011; Laakso et al., 1990; Vincent et al., 2003). Given that insulin-resistant rodents and humans show capillarity rarefaction, this highlights the importance of sufficient muscle capillarisation for insulin-mediated glucose disposal (Bonner et al., 2013; Chung et al., 2006; Gavin et al., 2005; Guo et al., 2012; Lillioja et al., 1987; Mårin et al., 1994). Furthermore, the angiotensin II receptor blocker losartan (Guo et al., 2012), which enhances

vasculature-induced tissue perfusion, also improves insulin sensitivity and increases microvascular density in skeletal muscles (Bonner et al., 2013; Chai et al., 2011; Kang et al., 2011; Guo et al., 2012).

Using $mVEGF^{-/-}$ mice, Bonner et al. (2013) showed an ∼60% decrease in capillary density in skeletal muscle. Moreover, KO mice lacking vascular endothelial growth factor present with reduced insulin-mediated glucose disposal (Bonner et al., 2013). Importantly, Bonner et al. observed that this reduction in insulin-stimulated glucose uptake in skeletal muscle was not associated with a reduction in intracellular insulin signalling (IRS-1, p85 and phosphorylated and total (p/t) Akt), suggesting that reduced insulin-stimulated muscle glucose uptake was caused by poor muscle perfusion (Bonner et al., 2013). Thus, it is hard to draw a complete picture of insulin signalling in skeletal muscle as it is possible that integrin-associated signalling could have been negatively affected in this $mVEGF^{-/-}$ rodent model (Bonner et al., 2013). Taken together, the studies above suggest that muscle capillary rarefaction and endothelial dysfunction are two avenues through which the remodelling of the ECM may contribute to insulin resistance in skeletal muscles (Williams et al., 2015).

## Integrins: important players in diet-induced insulin resistance

It seems clear that excessive ECM protein remodelling harms insulin sensitivity in skeletal muscle. However, it also appears possible that loss of function in key receptor proteins may also be implicated in insulin resistance. Of the two main integrin-mediated adhesion molecules, integrin $\alpha2\beta1$ is of particular interest as under stress conditions such as a HFD, it reveals an antiangiogenic and profibrotic nature, promoting increased collagen expression and reactive oxygen species production (Bedard & Krause, 2007; Kang et al., 2011; Langholz et al., 1995). Integrin $\alpha2\beta1$ deletion in mouse skeletal muscle leads to angiogenesis and cell proliferation *in vivo* (Kang et al., 2011; Zhang et al., 2008). In contrast, the collagen receptor integrin $\alpha1\beta1$ is proangiogenic and antifibrotic, with integrin $\alpha1$-null mesangial cells displaying increased collagen synthesis (Borza et al., 2012; Chen et al., 2004, 2007). Moreover, integrin $\alpha1\beta1$ deletion causes a reduction in angiogenesis and endothelial cell proliferation *in vivo* (Abair et al., 2008; Pozzi et al., 2000).

Zong et al. (2009) showed that deletion of muscle-specific integrin $\beta1$ in chow-fed mice led to a reduction in whole-body insulin-stimulated glucose uptake during a hyperinsulinaemic–euglycaemic clamp, a finding that was accompanied by a decrease in muscle glycogen synthesis and p-Akt$^{Ser473}$. This notion is demonstrated schematically in Fig. 2*B*.

Using mouse models lacking integrin $\alpha2\beta1$ ($itg\alpha2^{-/-}$) and integrin $\alpha1\beta1$ ($itg\alpha1^{-/-}$), Kang et al. (2011) showed that a HFD caused insulin resistance in both controls and integrin-null mice, yet glucose infusion rates (GIRs) were reduced in $itga1^{-/-}$ compared with control ($itga1^{+/+}$) mice in response to HFD treatment. In addition, GIRs were higher in the high-fat (HF)-fed $itg\alpha2^{-/-}$ mice when compared with HF-fed $itg\alpha2^{+/+}$ mice. Moreover, HF-fed $itg\alpha2^{-/-}$ skeletal muscles had increased IRS-1 and p-Akt expression compared to controls (Kang et al., 2011). Taken together, these data imply that HFD mice lacking integrin $\alpha2\beta1$ have improved insulin sensitivity, suggesting that integrin $\alpha2\beta1$ may play a role in the development of insulin resistance.

Interestingly, the HF-fed $itg\alpha2^{-/-}$ mice seemed to be protected against HFD-induced insulin resistance, associated with the presence of a normal collagen protein structure (Kang et al., 2011). Moreover, the decreased insulin resistance observed in an integrin $\alpha2\beta1$ null rodent may be explained by an enhanced vascularisation in HF-fed $itg\alpha2^{-/-}$ mice compared to controls as shown using immunostaining techniques (Kang et al., 2011), providing further evidence of the anti-angiogenic nature of integrin $\alpha2\beta1$.

The study by Kang et al. (2011) provides mechanistic insight into the potential link between muscle insulin resistance, decreased glucose disposal and increased collagen structure, which seems to be reversed in the absence of integrin $\alpha2\beta1$. Importantly, HF-fed rodents in which integrin $\alpha2\beta1$ was deleted had significantly greater insulin-stimulated glucose uptake compared with chow-fed $itg\alpha2^{+/+}$ mice. This study proposes a key role of integrins in insulin resistance and glucose uptake, but it does not explain how integrins, at a mechanistic level, connect to the cytoskeleton and exert their functions. The next section will review the key integrin-associated proteins along with the potential role these proteins play in ECM-cytoskeleton stabilisation and glucose uptake in skeletal muscle.

## ILK-deficient mice under HFD show improved insulin sensitivity and muscle capillarisation

Kang et al. (2016) used muscle-specific ILK-deficient (ILK$^{lox/lox}$HSAcre) mice to study ILK under diet-induced muscle insulin resistance. ILK$^{lox/lox}$HSAcre and WT mice (ILK$^{lox/lox}$) were fed a HFD or chow diet for 16 weeks (Kang et al., 2016), with validated insulin clamps being performed at 0 and 16 weeks. ILK deficiency did not alter body mass or body fat in the ILK$^{lox/lox}$HSAcre mice. In addition, glucose infusion rates and glucose disappearance ($R_d$) rates were not different between chow-fed groups, whereas both parameters were substantially higher in HF-fed ILK$^{lox/lox}$HSAcre compared

with HF-fed WT, suggesting greater insulin-mediated glucose response in the ILK-deficient mice (Kang et al., 2016). Glucose metabolic index ($R_g$) was greater in HF-fed WT compared with chow-fed WT, meaning the mice developed HFD-induced insulin resistance; however, the HFD-induced glucose metabolism impairment was absent in the HF-fed ILK$^{lox/lox}$HSAcre mice (Kang et al., 2016). Moreover, the same authors found that $R_g$ was not different between WT and ILK-deficient mice in adipose tissue, regardless of diet. These results suggest that, independently of adiposity, muscle-specific ILK deletion improves glucose intolerance despite diet-induced insulin resistance (Kang et al., 2016).

Interestingly, Wasserman's group found that improved insulin resistance was associated with an increase in CD31, a vascular/endothelial marker protein found in proximity to capillaries, implying increased signalling for local capillarisation in the HFD ILK-deficient mice (Kang et al., 2016). The increase of CD31 in HF-fed ILK$^{lox/lox}$HSAcre mice was also associated with a decrease in c-Jun N-terminal kinase, P38 and extracellular signal-regulated kinases 1 and 2, all of which are known to inhibit endothelial function and capillary proliferation, with an increase in insulin-dependant Akt phosphorylation (Kang et al., 2016). In sum, muscle-specific ILK-deficient mice showed increased muscle capillarisation and increased Akt activity, highlighting the importance of ILK in insulin perfusion and insulin signalling in skeletal muscle.

### ILK depletion impairs glucose uptake in muscles of adult mice in a non-pathological context

Elsewhere, a study by Hatem-Vaquero et al. (2017) used whole-body conditional ILK knockdown (cKD-ILK) mice to make comparisons with WT counterparts in response to a typical chow-fed diet with comparisons for glucose control measured by glucose (GTT) and insulin (ITT) tolerance test. These data showed elevated glycaemia and insulinaemia in cKD-ILK mice and increased homeostasis model assessment of insulin resistance (HOMA-IR; Hatem-Vaquero et al., 2017). A parallel *in vitro* experiment showed that GLUT4 expression and p-Akt$^{Ser473}$ were reduced in cKD-ILK tissues (Hatem-Vaquero et al., 2017). Taken together, these results suggest that ILK depletion in adult mice in a non-pathological context impairs GLUT4 expression and membrane presence, with subsequent reduction in peripheral glucose uptake and insulin sensitivity (Hatem-Vaquero et al., 2017). It is worth noting that these authors intended to perform this research on healthy, basal-fed adult mice, to avoid any upstream ILK pathological changes linked to ECM-associated proteins content, a phenomenon typically seen in type 2 diabetes

(Berria et al., 2006; Hatem-Vaquero et al., 2017; Kang et al., 2011; Pasarica et al., 2009; Williams et al., 2015). Overall, Hatem-Vaquero et al.'s (2017) results show differences in glucose homeostasis between the cCK-ILK and WT mice. This is contrary to the observations of Kang et al. (2016), where the non-inducible skeletal muscle-specific ILK KO mice showed no differences in glucose homeostasis compared to WT under basal chow-fed diet. This discrepancy may be explained by the different origin and settings of the mouse model used; Kang et al.'s (2016) KO mice were developed through a muscle-specific ILK deletion at birth, whereas Hatem-Vaquero el al.'s (2017) models of ILK depletion were induced in adulthood. That said, what both these studies seem to confirm is that ILK is an important effector in the integrin nexus that acts as a downstream regulator in diet-induced insulin resistance.

### The ILK–Rac1–cytoskeleton pathway is involved in insulin resistance

Insulin-resistant skeletal muscle and muscle cells in culture display altered actin remodelling. Sylow et al. (2013) demonstrated decreased Rac1 expression by 20% in soleus and 15% in EDL, along with reduced glucose uptake in mice feed a HFD for 12 weeks. Elsewhere Raun et al. (2018) showed that HFD Rac1 muscle-specific knockout (mKO) mice display reduced insulin-stimulated glucose transport in triceps, gastrocnemius and soleus muscles, suggesting that Rac1 ablation and HFD treatment combined negatively effects insulin-dependent glucose transport in muscle. These mouse studies are corroborated in human-based experimental trials showing that Rac1 signalling is impaired in insulin-resistant patients with T2D and in obese subjects (Sylow et al., 2013). ILK has been also shown to have a role in regulating actin filament rearrangement via PI3K–Rac1 (Gheyara et al., 2007; Qian et al., 2005). Indeed, ILK overexpression in CEF cells increased Rac1 activation, a finding completely reversed with the administration of the PI3K inhibitor LY294002 (10–20 $\mu$M), suggesting that Rac1 is implicated in ILK signalling in a PI3K-dependent manner (Qian et al., 2005). In addition, the same authors demonstrated that Rac1 inhibition in CEF cells inhibited the effects of ILK on active filament and cell migration (Qian et al., 2005). A similar link between ILK and Rac1 could therefore be speculated in skeletal muscle as well.

ILK depletion in *Caenorhabditis elegans* and *Drosophila melanogaster* leads to muscle detachment of focal adhesion sites (Alessi et al., 1997; Gheyara et al., 2007; Mackinnon et al., 2002; Zervas et al., 2001), while ILK KO mice presented severe muscular dystrophy and actin cytoskeleton restructuring and displacements of focal adhesion-related proteins, such as dystrophin and FAK (Boccafoschi et al., 2011; Hodges et al., 1997; Gheyara

et al., 2007). Taken together, these findings demonstrate that ILK is a possible actin cytoskeleton regulator sitting upstream of Rac1. However, Gheyara et al. (2007) did not investigate the metabolic consequences of ILK depletion. Thus, we speculate that glucose uptake may have decreased in the ILK KO model, given that ILK-depleted mice also showed a reduction in mass of skeletal muscle, a tissue that accounts for 80% of the total glucose metabolism under insulin stimulation (Bisht & Dey, 2008). The hypothesis is supported elsewhere (Hatem-Vaquero et al., 2017; Kang et al., 2016).

## Conclusion

The integrins and the actin cytoskeleton are suggested to have an important role in contraction- and insulin-stimulated glucose uptake in muscle. This functional link involves the ECM-associated integrin network, which might be required for all stimuli to be effective and for inducing glucose transport in skeletal muscle. It is worth noting that the tension-mediated pathway seems to operate alongside both contraction- and insulin-stimulated glucose uptake, rather than separately. While the interacting proteins involved in this regulatory process need further investigation, integrin-associated proteins and the actin cytoskeleton have potential roles. ECM remodelling occurs in mouse and human models of obesity and insulin-resistant and type 2 diabetes, which are linked, in part, to elevated systemic inflammation, which is in turn associated with pro-fibrotic mechanisms. The ECM involvement in obesity-induced insulin resistance may be explained by the evidence that ECM remodelling contributes to increased collagen formation, a process responsible for generating a physical barrier for insulin diffusion and glucose transport. Moreover, ECM remodelling causes distinct compositional changes in integrin signalling within the matrix, again a process known to contribute to reduced insulin and nutrient uptake in skeletal muscle. The research discussed in this review suggests that a stable linkage between the ECM and the cytoplasmic actin cytoskeleton is critical for effective glucose uptake in skeletal muscle in response to insulin-, contraction- and tension-mediated pathways. However, we cannot yet ascertain if the ECM–integrin axis operates independently of the traditional pathways, or merely works alongside these to assist effective glucose uptake. This research area is clearly still in its infancy with a number of key questions unanswered and requiring technological advances.

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

## Additional information

### Competing interests

None.

### Author contributions

F.D., R.M. and L.S. contributed to the conception of the manuscript. F.D. and R.M. contributed to drafting and writing the manuscript. L.S., V.B., N.T. and N.H. contributed to revising intellectual content of the manuscript. F.D. and R.M. had primary responsibility for final content. All authors have read and approved the final version of this manuscript and agree to be accountable for all aspects of the work in ensuring that questions related to the accuracy or integrity of any part of the work are appropriately investigated and resolved. All persons designated as authors qualify for authorship, and all those who qualify for authorship are listed.

### Funding

This research did not receive any specific grant from funding agencies in the public, commercial, or not-for-profit sectors.

### Keywords

actin cytoskeleton, ECM, ILK, insulin, insulin resistance, integrin, muscle contraction, Rac1

## Supporting information

Additional supporting information can be found online in the Supporting Information section at the end of the HTML view of the article. Supporting information files available:

**Peer Review History**

