## [Peer Review History · The Journal of Physiology]

Involvement of the extracellular matrix and integrin signalling proteins in skeletal muscle glucose uptake

Richard WA Mackenzie, Fulvia Draicchio, Volker Behrends, Neale Anthony Tillin, Nicholas Hurren, and Lykke Sylow
DOI: 10.1113/JP283039

Corresponding author(s): Richard Mackenzie (richard.mackenzie@roehampton.ac.uk)

The following individual(s) involved in review of this submission have agreed to reveal their identity: David H. Wasserman (Referee #2)

Review Timeline:

Submission Date:	28-Feb-2022
Editorial Decision:	12-Apr-2022
Revision Received:	21-Jul-2022
Accepted:	03-Aug-2022

Senior Editor: Ian Forsythe

Reviewing Editor: Javier Gonzalez

Transaction Report:

Dear Dr Mackenzie,

Re: JP-TR-2022-283039 "Are the extracellular matrix and integrin signalling proteins required for effective glucose transport in skeletal muscle?" by Richard WA Mackenzie, Fulvia Draicchio, Volker Behrends, Neale Anthony Tillin, Nicholas Hurren, and Lykke Sylow

Thank you for submitting your Topical Review to The Journal of Physiology. It has been assessed by a Reviewing Editor and by 2 expert referees and I am pleased to tell you that it is considered to be acceptable for publication following satisfactory revision.

The reports are copied at the end of this email. Please address all of the points and incorporate all requested revisions, or explain in your Response to Referees why a change has not been made.

NEW POLICY: In order to improve the transparency of its peer review process The Journal of Physiology publishes online as supporting information the peer review history of all articles accepted for publication. Readers will have access to decision letters, including all Editors' comments and referee reports, for each version of the manuscript and any author responses to peer review comments. Referees can decide whether or not they wish to be named on the peer review history document.

I hope you will find the comments helpful and have no difficulty in revising your manuscript within 4 weeks.

Your revised manuscript should be submitted online using the links in Author Tasks Link Not Available. This link is to the Corresponding Author's own account, if this will cause any problems when submitting the revised version please contact us.

You should upload:

- A Word file of the complete text (including any Tables);
- An Abstract Figure, (with accompanying Legend in the article file)
- Each figure as a separate, high quality, file;
- A full Response to Referees;
- A copy of the manuscript with the changes highlighted.
- Author profile. A short biography (no more than 100 words for one author or 150 words in total for two authors) and a portrait photograph of the two leading authors on the paper. These should be uploaded, clearly labelled, with the manuscript submission. Any standard image format for the photograph is acceptable, but the resolution should be at least 300 dpi and preferably more.

- A 'Cover Art' file for consideration as the Issue's cover image;
- Appropriate Supporting Information (Video, audio or data set https://jp.msubmit.net/cgi-bin/main.plex?form_type=display_requirements#supp).

To create your 'Response to Referees' copy all the reports, including any comments from the Senior and Reviewing Editors into a Word, or similar, file and respond to each point in colour or CAPITALS. Upload this when you submit your revision.

I look forward to receiving your revised submission.

Yours sincerely,

Ian D. Forsythe
Deputy Editor-in-Chief
The Journal of Physiology
<https://jp.msubmit.net>
<http://jp.physoc.org>
The Physiological Society
Hodgkin Huxley House
30 Farringdon Lane
London, EC1R 3AW
UK
<http://www.physoc.org>
<http://journals.physoc.org>

EDITOR COMMENTS

Reviewing Editor:

The present manuscript has been reviewed by two experts in the area. Both reviewers highlight strengths of the manuscript in relation to the topic that is covered. There are a number of points that do, however, need addressing, mainly in relation to the title and general structure/organisation. Please pay careful attention to these in revising this manuscript.

Senior Editor:

Thank you for this submission. As you see the referees raise many important issues which will require a very significant re-write (and response). As you are doing this please consider how you might shorten and re-organise your article, and provide succinct summaries if topics have been considered elsewhere recently.

During your revision, please re-write the abstract to provide more factual information about the science you are covering and avoid statements like "This review paper explores the current discussion of the role of ECM and actin-cytoskeleton in..." by directly stating your findings. And come to a clear conclusion in the final sentence of the abstract (please avoid concluding more work is required or is relevant to this or that therapy - that should be in the main text).

Please consider how to make your abstract figure simple and self explanatory.

I look forward to receiving your revised MS.

REFeree COMMENTS

Referee #1:

The manuscript by Draicchio et al is a topical review entitled "Are the extracellular matrix and integrin signalling proteins required for effective glucose transport in skeletal muscle?"

The proposed review is original and in the current wave of many reviews on glucose transport and metabolism this is an angle which has not been explored recently. With this in mind the review is topical and justified. The authors have chosen to review only aspect related to skeletal muscle glucose transport and linking it to whole body glucose homeostasis justifying their choice by the fact the skeletal muscle is the main tissue responsible for glucose disposal.

Below are listed the points which would need further attention:

1. The title: I would suggest changing it as for most this review reports results from mouse models of putative candidate interactors knockdowns or overexpression while monitoring signalling cascades leading to glucose transport regulation and glucose uptake or measuring whole body responses to diet interventions inducing insulin resistant state. Although all these processes are interlinked and inevitably will affect glucose transport or a due to alterations in glucose transport, the reviewed evidence do not address the question stated in the title.
2. Overall, the language and writing style are very good. However, I would suggest reviewing the structure of the whole manuscript as there are parts which feel out of place and logically should have been discussed much earlier. An example of this can be seen on page 20 of the manuscript:

"This study proposes a key role of integrins in insulin resistance and glucose uptake, however they do not explain how integrins, at a mechanistic level, connect to the cytoskeleton and exert their functions. Integrins lack endogenous enzymatic activity, so they are believed to form focal adhesions (FAs), composed of complex groups of downstream signalling molecules and proteins to provide a link to actin and microtubule cytoskeletons (Hynes, 2002; Schober et al., 2007; Clemente et al., 2012). The next section will review the key integrin-associated proteins along with the potential role these protein's play in ECM cytoskeleton stabilisation and glucose uptake in skeletal muscle."

In my opinion this should have been introduced much earlier when describing integrins in page 10.

Similarly, the last paragraph on page 24 is re-visiting the Gheyara et al 2007 study feels out of place, and it would have been good to discuss it earlier when beginning to talk about the different KO models of ILK as this model precedes the muscle KO and conditional KO models described on pages 20 to 22.

3. In the general introduction a point is made that studying less explored pathways such as ECM could be a potential therapeutic target for people with insulin resistance and type 2 diabetes. This is fine as it is the basis of the review, however few sentences further down it is stated that "integrins, and their downstream effectors, are emerging as key players in skeletal muscle insulin action and glucose uptake" citing the review by Williams et al 2015. Maybe this statement should be slightly attenuated as even the evidence presented in the current manuscript taken together with the evidence in the excellent review by Williams et al. are far from pointing at the ECM/integrin pathways as a key player. They are undoubtedly contributing and could be important for the understanding of the development of insulin resistant state, but I would suggest toning down this statement. Similarly in the conclusion (page 25) it is stated that the role of integrins in insulin-stimulated

glucose uptake is well established - again I would suggest revising.

4. The section "Canonical pathways..." - given the amount of reviews on muscle glucose uptake, which have been published in the last couple of years, some by the co-authors of this manuscript, I would advise referring to these excellent reviews and just summarising what is the current understanding of how insulin and contraction regulates glucose uptake in skeletal muscle. The authors have attempted to do this, but it is rather confusing and, in some places, imprecise. For example, GLUT4 translocation in response to insulin and contraction is not mentioned until the end of second paragraph well after talking about phosphoproteomics which is surprising. Page 6 the references for the requirement for Akt1/2 in insulin stimulated GLUT4 translocation in muscle are given as Wang et al., 1999; Al-Khalili et al., 2006. The first of these references was unknown to me so I have looked in the reference list but it is not listed so I cannot comment further; the second one Al-Khalili et al, I would not have used a reference to support this statement. There are many other authors and work that comes to mind who have done the essential demonstration of the involvement of PKB/Akt in insulin stimulated GLUT4 translocation in skeletal muscle Brozinick and Birnbaum 1998 and Lund et al., 1998 come to mind.

5. Overall, and this is something which is especially valid for this same section discussed in point 4, citing few reviews is fine in a review paper, but by referencing mainly review articles we seem to be losing the essence of what the field has achieved, and we are not crediting anymore the real pioneers in the field. Beside the example I just provided above another example is that the convergence of insulin and contraction signalling pathways has not been described first by (Kim et al., 2021; Richter et al., 2021; Ihlemann et al., 2000) as it currently reads. I think it is important to credit the original papers and authors. I acknowledge that the authors of this current manuscript have contributed to some of those excellent reviews which are cited but citing the same reviews over and over is perhaps too much. I would advise either summarising everything and referring the reader to the reviews or putting the facts right and crediting the original authors.

6. Figure 2A and B needs some attention as it is not obvious what are the changes especially that in Fig 2B InsR and Akt are left with the phospho groups, so it is difficult to see the suppression of the signalling. Also, according to most of the reports the proximal steps of the insulin signalling cascade are not affected in people with insulin resistance. So it is difficult to see in the proposed model how the integrin signalling pathway would fit in this model if it is affecting these proximal steps as illustrated in Figure 2. Could it be an action essentially via Rac1 and actin remodelling? If the model stays as it is Fig.2A TBC1D4 should be with phospho residues to show the inactivation and then in Fig. 2B TBC1D4 should be still associated with the GSVs and not phosphorylated. I suggest removing the dotted arrow in 2B.

7. The current review up to page 17 for most recapitulates what is already reviewed by Williams et al 2015 and the authors themselves are citing this review many times to support their statements. A suggestion would be to summarise many of these points by referring the readers to this previous review and then define what is the angle the authors are taking that makes this review different.

8. ILK KO models description and comparisons page 22 and 23. - The authors report that in the cKD-ILK model Hatem-Vaquero et al report that ILK depletion impairs GLUT4 translocation. This is not factually correct as what Hatem-Vaquero et al observe is that there is general decrease in the GLUT4 total protein content not in the translocation to the PM. It will be also helpful if the authors mention that the cKD-ILK model is a whole-body knockdown as this is not mentioned at all, but it is important to mention with regards to comparisons with other models, especially the muscle-specific ILK KO model by Kang et al 2016.

9. When describing a potential link between ILK1-PI3K-Rac1 on page 23 it will be advisable to report that the cited studies have reported this link in other tissues, not skeletal muscle. Therefore, the link that the authors propose for the action of ILK in skeletal muscle although plausible is speculative rather than based on evidence. I think it is fine to have some speculative element or a proposed future avenue of investigation even if this is a review article but perhaps it will be good to bring together other of those elements which are slightly scattered around. For example, the authors introduce at the beginning the proteoglycans and the dystrophin-glycoprotein complex making the points that they are essential for maintenance of muscle structure and could be important for transmitting signals from the ECM to the intracellular signalling pathways leading to actin remodelling. However, there is no further reference to these complexes in the review therefore it is questionable why they are mentioned in the first place and whether they have (or not) a role to play. As mentioned above it would be good to have at the end a section discussing the potential avenues to be explored and what the authors think is the role for the ECM in maintenance of glucose uptake. This is covered to a certain degree in the conclusion, but in light of restructuring some of the elements of this review maybe it could be further elaborated.

10. Minor points which need attention:

- Abstract: "...with recent developments suggesting convergence between these two previously separate processes." The convergence of the insulin and contraction pathways leading to GLUT4 translocation is not a recent development it is 20+ years old.

- Page 4 - top of the page the sentence is not complete.

"...supports the notion that actin cytoskeleton, extracellular matrix (Richardson et al., 2005;..."

- Page 5 : cytoplasmic should be cytoplasm? "...Upon stimulation, GLUT4s translocate from GLUT storage vesicles (GSV) in the cytoplasmic to the cell surface to facilitate glucose uptake across the sarcolemma (Sylov et al.,..."

- Page 6 if details of the insulin and contraction signalling pathways leading to GLUT4 translocation are given then TBC1D1 should be referred to as well with respect to the contraction pathway.
- Page 10 - top of the page first sentence needs attention. I don't think that it is the ECM which is "responsible for intracellular communication". Would it be that the ECM communicate with cells via the integrins?
- I would suggest adding Rac1 in Figure 1 to provide to link it to the very good paragraph on page 11 regarding Rac1 connection to Actin/Talin/Vinculin and connect it to the rest of the review. Reference to this figure 1 on page 11 could be helpful to the reader.

Referee #2:

The paper by Draicchio and colleagues highlights a fundamental signaling axis common to all cells. Despite the essential nature of integrins, focal adhesions and their interaction with the extracellular environment there is little known about how they affect metabolic physiology. The muscle is the focus here because of its role in insulin- and contraction-stimulated glucose uptake. I found the paper to be clearly organized and written. The value of this review is that it highlights an area that requires considerably more attention. Most of the work in this field has been in developmental, injury, and cancer biology.

Wording the title as a question is a good approach. However, I suggest a couple changes. One is I suggest changing the word "required". I suspect that integrin signaling proteins as an aggregate are "required" for cells to live. Consider changing to something like "necessary for the regulation". The authors make a good case that disrupting ECM-integrin signaling alters glucose uptake by muscle. It is impossible to affect glucose uptake without increasing glucose transport flux. I believe what the authors really want to ask whether ECM-integrin signaling is coupled to GLUT4 translocation? This fits with the emphasis on GTPases and cytoskeleton in the review. I suggest the authors consider more specific messaging.

This review is novel because most scientists studying the cell adhesion examine the biochemistry and biophysics of the basement membrane, integrin receptors, and the cell cortical region without coupling to physiology. The authors ask what is the role of ECM-integrin signaling on glucose transport? The integrin link to GTPases and cytoskeleton are well established. Once one acknowledges the role of the cytoskeleton and interacting proteins there are pathways besides glucose translocation that are likely affected (organelle structure, fatty acid trafficking). It is too much to get into all potential affected processes. However, it may be good to emphasize that once you've messed with the cytoskeleton all sorts of processes are potentially affected.

Again, it is too much to discuss in detail. But the schemes that the authors have included and ones that we have used are minimalist, only including the proteins necessary to make a discussion. It would be fair to say that the adhesion has many proteins beyond the scope of the review (maybe 60 or more) and that the number of interactions/potential interactions are vast. These are beyond the scope of the review, but perhaps it should be clarified that the schemes are not naïve so much as minimal models and that the full system includes many more players and interactions that are well beyond those mentioned.

Authors might consider playing up the challenges of studying ECM-integrins in explaining why so little is known. For example, the physiological role of integrins is very difficult to address in isolated cell systems. This is because it is difficult to recreate the physiological ECM and cell-cell junctions in media and to maintain a similar expression profile when focal adhesions are disrupted. In practice, these barriers have been an obstacle to stem cell differentiation and proliferation and regenerative medicine. There is a need therefore for good model systems. A second reason so little is known about the cell adhesion is not for lack of importance but rather because many of the key proteins don't seem to have catalytic activity (integrin $\beta 1$ subunit, IPP complex, NCK). This has made this more of a biophysics area of study than about the biochemistry of ser/thr phospho "signaling".

Page 5 - There is no shortage of reviews on GLUT4 trafficking. Last sentence of the big paragraph cites 5 reviews on GLUT4 trafficking that all come from Copenhagen. Probably two of the more recent reviews from the Copenhagen group would suffice with the addition of the recent work from another group just to be balanced. No denying the expertise in Copenhagen. But with as many groups that work in this area it would be fair to acknowledge other perspectives. Might consider a review by Debbie Thurmond.

p. 7, l. 8-9 - I don't know if it's exactly correct to say "directly downstream". There is divergence and parallel events that may be occurring in this cascade. Good enough to delete "directly". The subsequent lines about the complexity of the ECM are very good. Perhaps a couple lines about the cells that secrete the ECM proteins and what stimulates them to do so would give some context.

p. 7, last sentence - Specify which Col or if you are referring to all Cols turn sentence into plural. Collagens are....

p. 12-misspelling of Brozinick in the 2007 call out.

p. 12-second paragraph - The turnover of ECM over time is slow relative to changes in glucose/insulin. Is the ECM-integrin axis an independent regulator of glucose transport or does it modify the sensitivity to insulin, other tyrosine kinases, etc? In

response to a meal, I would expect glucose and insulin to increase but I would not expect a change in ECM. On the other hand, Mandarino shows there can be changes in ECM proteins rather quickly with lipid infusion.

p. 13, last line - ILK recruits the downstream effectors alpha-actinin and Rac1 through parvin. Is "recruits" the correct word? As far as I know the amount of these proteins remains unchanged by ILK, at least in an acute sense.

p. 16, discussion of Kang HA paper. The authors correctly address all aspects of this paper. The transition between sentences implies to me that there is an inconsistency in the results. The authors' interpretation is in retrospect questionable (depletion of the glycocalyx may also explain findings) but the results are internally consistent.

p. 19, itg^{-/-} decreases GIR. This as a standalone is confusing. Authors should specify this decrease is due to a liver effect (i.e., nothing to do with muscle).

p. 19, last paragraph - One thing to consider mentioning is that the implication of itg^{-/-} increasing insulin action without decreasing ECM protein suggests that the effect of ECM is receptor-mediated and not a result of a physical barrier.

p.20, last paragraph - No need to specify ICv. I don't think the v was supposed to stand for validated. I think it was supposed to stand for Vanderbilt. Vanderbilt is just so proud of its clamp. Fortunately, it was a short-lived phase, I hope.

p. 22 - This looks like it should be combined with the Discussion at the bottom of page 13. I think it makes sense to put them together. If there is something that should be distinguishing these two sections please clarify so it doesn't appear redundant. The explanation of the ILK KO vs cKD ILK are reasonable. There are other possible explanations. Studies in the liver by Trefts et al. 2019 showed using cKD of ILK that there is the transient appearance of an injured liver that becomes normalized over time (6 to 18 wks). The time courses in Figures 1 and 2 in the cKD paper also suggest a convergence over time. To the extent that the response in liver can be compared to muscle there is evidence that the response in conditional KD is time-dependent. The other major issue is that the Hatem-Vaquero is whole body cKD. Whereas Kang (and Trefts) was tissue specific.

Heading on page 22. Not sure what is meant by "non-pathological context." The studies describe HFD mice and Type 2 diabetes. By "basal-fed" do you mean standard chow?

p. 23, l. 4 - "ILK is an important kinase". I suggest rephrasing. There is a kinase domain and kinase activity in a test tube. It is contentious as to whether there is kinase activity in cells or whole animal. I believe the consensus is that the ILK kinase domain is a binding site for parvin and some other proteins. ILK is an important protein but probably not for kinase activity.

There are two issues that may be relevant to the discussion that the authors should consider touching on.

PINCH is shown in the graph. The initial view going back several decades was that the IPP would affect glucose transport by PINCH-NICK inhibition of insulin signaling. I agree with the case made in the Review. It is more likely to be GTPase and actin-cytoskeleton related. However, I suggest the authors leave the PINCH option in play.

The integrin receptor field is divided (or doesn't really know) whether the more important activator of integrin signaling is by ligand binding or physical force applied to the extracellular domain of the integrin receptor. The latter could be relevant in a condition like muscle contraction and would be an interesting point to acknowledge.

David Wasserman

REQUIRED ITEMS:

- Please include an Abstract Figure. The Abstract Figure is a piece of artwork designed to give readers an immediate understanding of the Review Article and should summarise the main conclusions. If possible, the image should be easily 'readable' from left to right or top to bottom. It should show the physiological relevance of the Review so readers can assess the importance and content of the article. Abstract Figures should not merely recapitulate other figures in the Review. Please try to keep the diagram as simple as possible and without superfluous information that may distract from the main conclusion of the Review. Abstract Figures must be provided by authors no later than the revised manuscript stage and should be uploaded as a separate file during online submission labelled as File Type 'Abstract Figure'. Please ensure that you include the figure legend in the main article file. All Abstract Figures will be sent to a professional illustrator for redrawing and you may be asked to approve the redrawn figure before your paper is accepted.

- Your MS must include a complete "Additional information section" with the following 4 headings and content:

Competing Interests: A statement regarding competing interests. If there are no competing interests, a statement to this effect must be included. All authors should disclose any conflict of interest in accordance with journal policy.

Author contributions: Each author should take responsibility for a particular section of the study and have contributed to writing the paper. Acquisition of funding, administrative support or the collection of data alone does not justify authorship;

these contributions to the study should be listed in the Acknowledgements. Additional information such as 'X and Y have contributed equally to this work' may be added as a footnote on the title page.

It must be stated that all authors approved the final version of the manuscript and that all persons designated as authors qualify for authorship, and all those who qualify for authorship are listed.

Funding: Authors must indicate all sources of funding, including grant numbers. If authors have not received funding, this must be stated.

It is the responsibility of authors funded by RCUK to adhere to their policy regarding funding sources and underlying research material. The policy requires funding information to be included within the acknowledgement section of a paper. Guidance on how to acknowledge funding information is provided by the Research Information Network. The policy also requires all research papers, if applicable, to include a statement on how any underlying research materials, such as data, samples or models, can be accessed. However, the policy does not require that the data must be made open. If there are considered to be good or compelling reasons to protect access to the data, for example commercial confidentiality or legitimate sensitivities around data derived from potentially identifiable human participants, these should be included in the statement.

Acknowledgements: Acknowledgements should be the minimum consistent with courtesy. The wording of acknowledgements of scientific assistance or advice must have been seen and approved by the persons concerned. This section should not include details of funding.

- Author profile(s) must be uploaded via the submission form. Authors should submit a short biography (no more than 100 words for one author or 150 words in total for two authors) and a portrait photograph of the two leading authors on the paper. These should be uploaded, clearly labelled, with the manuscript submission. Any standard image format for the photograph is acceptable, but the resolution should be at least 300 dpi and preferably more. A group photograph of all authors is also acceptable, providing the biography for the whole group does not exceed 150 words.

END OF COMMENTS

Confidential Review

28-Feb-2022

REFeree COMMENTS

Referee #1:

The manuscript by Draicchio et al is a topical review entitled "Are the extracellular matrix and integrin signalling proteins required for effective glucose transport in skeletal muscle?"

The proposed review is original and in the current wave of many reviews on glucose transport and metabolism this is an angle which has not been explored recently. With this in mind the review is topical and justified. The authors have chosen to review only aspect related to skeletal muscle glucose transport and linking it to whole body glucose homeostasis justifying their choice by the fact the skeletal muscle is the main tissue responsible for glucose disposal.

Below are listed the points which would need further attention:

1. The title: I would suggest changing it as for most this review reports results from mouse models of putative candidate interactors knockdowns or overexpression while monitoring signalling cascades leading to glucose transport regulation and glucose uptake or measuring whole body responses to diet interventions inducing insulin resistant state. Although all these processes are interlinked and inevitably will affect glucose transport or a due to alterations in glucose transport, the reviewed evidence do not address the question stated in the title.

This has been changed the title to better reflect the main topic of this paper.

2. Overall, the language and writing style are very good. However, I would suggest reviewing the structure of the whole manuscript as there are parts which feel out of place and logically should have been discussed much earlier. An example of this can be seen on page 20 of the manuscript:

"This study proposes a key role of integrins in insulin resistance and glucose uptake, however they do not explain how integrins, at a mechanistic level, connect to the cytoskeleton and exert their functions. Integrins lack endogenous enzymatic activity, so they are believed to form focal adhesions (FAs), composed of complex groups of downstream signalling molecules and proteins to provide a link to actin and microtubule cytoskeletons (Hynes, 2002; Schober et al., 2007; Clemente et al., 2012). The next section will review the key integrin-associated proteins along with the potential role these protein's play in ECM cytoskeleton stabilisation and glucose uptake in skeletal muscle."

In my opinion this should have been introduced much earlier when describing integrins in page 10.

Similarly, the last paragraph on page 24 is re-visiting the Gheyara et al 2007 study feels out of place, and it would have been good to discuss it earlier when beginning to talk about the different KO models of ILK as this model precedes the muscle KO and conditional KO models described on pages 20 to 22.

We have moved the section from page 20 to page 10, as you suggest.

3. In the general introduction a point is made that studying less explored pathways such as ECM could be a potential therapeutic target for people with insulin resistance and type 2 diabetes. This is fine as it is the basis of the review, however few sentences further down it is stated that "integrins, and their downstream effectors, are emerging as key players in skeletal muscle insulin action and glucose uptake" citing the review by Williams et al 2015. Maybe this statement should be slightly attenuated as even the evidence presented in the current manuscript taken together with the evidence in the excellent review by Williams et al. are far from pointing at the ECM/integrin pathways as a key player. They are undoubtedly contributing and could be important for the understanding of the development of insulin resistant state, but I would suggest toning down this statement. Similarly in the conclusion (page 25) it is stated that the role of integrins in insulin-stimulated glucose uptake is well established - again I would suggest revising.

Thank you, we think this is a fair observation. We have attenuated the statements accordingly.

4. -The section "Canonical pathways..." - given the amount of reviews on muscle glucose uptake, which have been published in the last couple of years, some by the co-authors of this manuscript, I would advise referring to these excellent reviews and just summarising what is the current understanding of how insulin and contraction regulates glucose uptake in skeletal muscle. The authors have attempted to do this, but it is rather confusing and, in some places, imprecise. For example, GLUT4 translocation in response to insulin and contraction is not mentioned until the end of second paragraph well after talking about phosphoproteomics which is surprising.

-Page 6 the references for the requirement for Akt1/2 in insulin stimulated GLUT4 translocation in muscle are given as Wang et al., 1999; Al-Khalili et al., 2006. The first of these references was unknown to me so I have looked in the reference list but it is not listed so I cannot comment further; the second one Al-Khalili et al, I would not have used a reference to support this statement. There are many other authors and work that comes to mind who have done the essential demonstration of the involvement of PKB/Akt in insulin stimulated GLUT4 translocation in skeletal muscle Brozinick and Birnbaum 1998 and Lund et al., 1998 come to mind.

Thank you for your suggestions, we have updated the references at page 6 to better support the statement. About the section starting with "Canonical pathways" please see the answer below.

5. Overall, and this is something which is especially valid for this same section discussed in point 4, citing few reviews is fine in a review paper, but by referencing mainly review articles we seem to be losing the essence of what the field has achieved, and we are not crediting anymore the real pioneers in the field. Beside the example I just provided above another example is that the convergence of insulin and contraction signalling pathways has not been described first by (Kim et al., 2021; Richter et al., 2021; Ihlemann et al., 2000) as it currently reads. I think it is important to credit the original papers and authors. I acknowledge that the authors of this current manuscript have contributed to some of those excellent reviews which are cited but citing the same reviews over and over is perhaps too much. I would advise either summarising everything and referring the reader to the reviews or putting the facts right and crediting the original authors.

Thank you for your suggestion. We think to have already summarized the topics enough, as it is crucial for the aim of this review to give the reader at least basic info about the mechanisms of the

traditional pathways that lead to GLUT4 translocation. However, I have added the original papers in the references, to let the reader know the original sources.

6. Figure 2A and B needs some attention as it is not obvious what are the changes especially that in Fig 2B InsR and Akt are left with the phospho groups, so it is difficult to see the suppression of the signalling. Also, according to most of the reports the proximal steps of the insulin signalling cascade are not affected in people with insulin resistance. So it is difficult to see in the proposed model how the integrin signalling pathway would fit in this model if it is affecting these proximal steps as illustrated in Figure 2. Could it be an action essentially via Rac1 and actin remodelling? If the model stays as it is Fig.2A TBC1D4 should be with phospho residues to show the inactivation and then in Fig. 2B TBC1D4 should be still associated with the GSVs and not phosphorylated. I suggest removing the dotted arrow in 2B.

We have amended Figures 2A and 2B accordingly. It's not clear Rac1 is involved at this stage, as proposed by the reviewer. Specifically, the following amendments have been completed, as suggested by the reviewer (Fig.2A TBC1D4 should be with phospho residues to show the inactivation and then in Fig. 2B TBC1D4 should be still associated with the GSVs and not phosphorylated. I suggest removing the dotted arrow in 2B).

7. The current review up to page 17 for most recapitulates what is already reviewed by Williams et al 2015 and the authors themselves are citing this review many times to support their statements. A suggestion would be to summarise many of these points by referring the readers to this previous review and then define what is the angle the authors are taking that makes this review different.

We thank the reviewer for this point and on face value, we were tempted to make the changes suggested here. On reflection, however, we felt that this area is still very new and felt that highlighting important points (as we see them) within the current review paper makes for a more informed reading experience, rather than have the reader move to a separate paper for key points.

Williams, A. S., Kang, L., & Wasserman, D. H. (2015). *The extracellular matrix and insulin resistance*. *Trends in Endocrinology & Metabolism*, 26(7), 357–366. <https://doi.org/10.1016/j.tem.2015.05.006>

8. ILK KO models description and comparisons page 22 and 23. - The authors report that in the cKD-ILK model Hatem-Vaquero et al report that ILK depletion impairs GLUT4 translocation. This is not factually correct as what Hatem-Vaquero et al observe is that there is general decrease in the GLUT4 total protein content not in the translocation to the PM. It will be also helpful if the authors mention that the cKD-ILK model is a whole-body knockdown as this is not mentioned at all, but it is important to mention with regards to comparisons with other models, especially the muscle-specific ILK KO model by Kang et al 2016.

Thank you to have spotted these imprecisions, we have amended the section accordingly.

9. When describing a potential link between ILK1-PI3K-Rac1 on page 23 it will be advisable to report that the cited studies have reported this link in other tissues, not skeletal muscle. Therefore, the link that the authors propose for the action of ILK in skeletal muscle although plausible is speculative rather than based on evidence. I think it is fine to have some speculative element or a proposed future avenue of investigation even if this is a review article but perhaps it

will be good to bring together other of those elements which are slightly scattered around. For example, the authors introduce at the beginning the proteoglycans and the dystrophin-glycoprotein complex making the points that they are essential for maintenance of muscle structure and could be important for transmitting signals from the ECM to the intracellular signalling pathways leading to actin remodelling. However, there is no further reference to these complexes in the review therefore it is questionable why they are mentioned in the first place and whether they have (or not) a role to play. As mentioned above it would be good to have at the end a section discussing the potential avenues to be explored and what the authors think is the role for the ECM in maintenance of glucose uptake. This is covered to a certain degree in the conclusion, but in light of restructuring some of the elements of this review maybe it could be further elaborated.

Thank you for this observation. Discussing further the role of the proteoglycans and the dystrophin-glycoprotein complex is not the scope of this review, thus we have reduced the section at pages 8-9.

10. Minor points which need attention:

- **Abstract:** "...with recent developments suggesting convergence between these two previously separate processes." The convergence of the insulin and contraction pathways leading to GLUT4 translocation is not a recent development it is 20+ years old.

We have amended the line accordingly.

- **Page 4 - top of the page** the sentence is not complete.

"...supports the notion that actin cytoskeleton, extracellular matrix (Richardson et al., 2005;..."

Thank you, we have completed the sentence.

- **Page 5 :** cytoplasmic should be cytoplasm? "...Upon stimulation, GLUT4s translocate from GLUT storage vesicles (GSV) in the cytoplasmic to the cell surface to facilitate glucose uptake across the sarcolemma (Sylov et al.,..."

Amended, thank you to have spotted this imprecision.

- **Page 6** if details of the insulin and contraction signalling pathways leading to GLUT4 translocation are given then TBC1D1 should be referred to as well with respect to the contraction pathway.

- **Page 10 - top of the page** first sentence needs attention. I don't think that it is the ECM which is "responsible for intracellular communication". Would it be that the ECM communicate with cells via the integrins?

Amended accordingly, thank you.

- I would suggest adding Rac1 in Figure 1 to provide to link it to the very good paragraph on page 11 regarding Rac1 connection to Actin/Talin/Vinculin and connect it to the rest of the review. Reference to this figure 1 on page 11 could be helpful to the reader.

Thank you – this has been added as had a reference to Figure 1. on page 11.

Referee #2:

The paper by Draicchio and colleagues highlights a fundamental signaling axis common to all cells. Despite the essential nature of integrins, focal adhesions and their interaction with the extracellular environment there is little known about how they affect metabolic physiology. The muscle is the focus here because of its role in insulin- and contraction-stimulated glucose uptake. I found the paper to be clearly organized and written. The value of this review is that it highlights an area that requires considerably more attention. Most of the work in this field has been in developmental, injury, and cancer biology.

Wording the title as a question is a good approach. However, I suggest a couple changes. One is I suggest changing the word "required". I suspect that integrin signaling proteins as an aggregate are "required" for cells to live. Consider changing to something like "necessary for the regulation". The authors make a good case that disrupting ECM-integrin signaling alters glucose uptake by muscle. It is impossible to affect glucose uptake without increasing glucose transport flux. I believe what the authors really want to ask whether ECM-integrin signaling is coupled to GLUT4 translocation? This fits with the emphasis on GTPases and cytoskeleton in the review. I suggest the authors consider more specific messaging.

Thank you for your helpful suggestions. We have changed the title in "Involvement of the extracellular matrix and integrin signalling proteins in skeletal muscle glucose uptake", for a more specific messaging.

This review is novel because most scientists studying the cell adhesion examine the biochemistry and biophysics of the basement membrane, integrin receptors, and the cell cortical region without coupling to physiology. The authors ask what is the role of ECM-integrin signaling on glucose transport? The integrin link to GTPases and cytoskeleton are well established. Once one acknowledges the role of the cytoskeleton and interacting proteins there are pathways besides glucose translocation that are likely affected (organelle structure, fatty acid trafficking). It is too much to get into all potential affected processes. However, it may be good to emphasize that once you've messed with the cytoskeleton all sorts of processes are potentially affected.

Again, it is too much to discuss in detail. But the schemes that the authors have included and ones that we have used are minimalist, only including the proteins necessary to make a discussion. It would be fair to say that the adhesion has many proteins beyond the scope of the review (maybe 60 or more) and that the number of interactions/potential interactions are vast. These are beyond the scope of the review, but perhaps it should be clarified that the schemes are not naïve so much as minimal models and that the full system includes many more players and interactions that are

well beyond those mentioned.

Authors might consider playing up the challenges of studying ECM-integrins in explaining why so little is known. For example, the physiological role of integrins is very difficult to address in isolated cell systems. This is because it is difficult to recreate the physiological ECM and cell-cell junctions in media and to maintain a similar expression profile when focal adhesions are disrupted. In practice, these barriers have been an obstacle to stem cell differentiation and proliferation and regenerative medicine. There is a need therefore for good model systems. A second reason so little is known about the cell adhesion is not for lack of importance but rather because many of the key proteins don't seem to have catalytic activity (integrin $\beta 1$ subunit, IPP complex, NCK). This has made this more of a biophysics area of study than about the biochemistry of ser/thr phospho "signaling".

This is a really good comment, thank you. We have added this on page 12, in the "*Integrin-associated proteins and the actin cytoskeleton – implications for glucose uptake*" section.

Page 5 - There is no shortage of reviews on GLUT4 trafficking. Last sentence of the big paragraph cites 5 reviews on GLUT4 trafficking that all come from Copenhagen. Probably two of the more recent reviews from the Copenhagen group would suffice with the addition of the recent work from another group just to be balanced. No denying the expertise in Copenhagen. But with as many groups that work in this area it would be fair to acknowledge other perspectives. Might consider a review by Debbie Thurmond.

This is a fair suggestion, thank you. Thurmond's work is indeed very interesting and we have added two papers from this research group.

p. 7, l. 8-9 - I don't know if it's exactly correct to say "directly downstream". There is divergence and parallel events that may be occurring in this cascade. Good enough to delete "directly". The subsequent lines about the complexity of the ECM are very good. Perhaps a couple lines about the cells that secrete the ECM proteins and what stimulates them to do so would give some context.

p. 7, last sentence - Specify which Col or if you are referring to all Cols turn sentence into plural. Collagens are....

Thank you, we have made both corrections on page 7.

p. 12-misspelling of Brozinick in the 2007 call out.

Amended, thank you.

p. 12-second paragraph - The turnover of ECM over time is slow relative to changes in glucose/insulin. Is the ECM-integrin axis an independent regulator of glucose transport or does it modify the sensitivity to insulin, other tyrosine kinases, etc? In response to a meal, I would expect glucose and insulin to increase but I would not expect a change in ECM. On the other hand, Mandarino shows there can be changes in ECM proteins rather quickly with lipid infusion.

This is a fantastic point and the very reason we enjoy the review process. I often go back and forth with this point given that some papers show changes in total intracellular protein signalling in response to a rather short intervention. Not too dissimilar to the Mandarino example above.

My guess, without much data to support this is that ECM-integrin axis potentially modifies insulin sensitivity (and contraction mediated glucose uptake), rather than acting independently; yet over time, the incorporation of some proteins from the ECM-integrin axis has developed. This is of course merely speculation. Yet we see some evidence of this with Rac1 interacting with integrin-talin-vinculin (page 11) and with examples of downstream integrin substrates include integrin-linked kinase (ILK), focal adhesion kinase (FAK) (Hynes, 2002; Schober *et al.*, 2007; Draicchio *et al.*, 2020), Akt, and Rac1 (Williams *et al.*, 2015; Draicchio *et al.*, 2020). But again, this could be a chicken and egg suggestion. We have made a note of this important point within the paper. Thank you again for the opportunity to respond to an interesting point.

p. 13, last line - ILK recruits the downstream effectors alpha-actinin and Rac1 through parvin. Is "recruits" the correct word? As far as I know the amount of these proteins remains unchanged by ILK, at least in an acute sense.

Thanks to have raised this observation. ILK is the central protein of the IPP complex and, from literature, its main role is often as a "bridge" between one and another protein, allowing cascade of events happening; as such, it is common to see it as a key molecule that recruits downstream effectors.

p. 16, discussion of Kang HA paper. The authors correctly address all aspects of this paper. The transition between sentences implies to me that there is an inconsistency in the results. The authors' interpretation is in retrospect questionable (depletion of the glycocalyx may also explain findings) but the results are internally consistent.

We checked the section related to Kang HA paper but we did not identify any incongruity / inconsistency. Could we have more clarification on this?

p. 19, itg-/- decreases GIR. This as a standalone is confusing. Authors should specify this decrease is due to a liver effect (i.e., nothing to do with muscle).

p. 19, last paragraph - One thing to consider mentioning is that the implication of itg-/- increasing insulin action without decreasing ECM protein suggests that the effect of ECM is receptor-mediated and not a result of a physical barrier.

p.20, last paragraph - No need to specify ICv. I don't think the v was supposed to stand for validated. I think it was supposed to stand for Vanderbilt. Vanderbilt is just so proud of its clump. Fortunately, it was a short-lived phase, I hope.

We have amended the sentence accordingly.

p. 22 - This looks like it should be combined with the Discussion at the bottom of page 13. I think it makes sense to put them together. If there is something that should be distinguishing these two

sections please clarify so it doesn't appear redundant. The explanation of the ILK KO vs cKD ILK are reasonable. There are other possible explanations. Studies in the liver by Trefts et al. 2019 showed using cKD of ILK that there is the transient appearance of an injured liver that becomes normalized over time (6 to 18 wks). The time courses in Figures 1 and 2 in the cKD paper also suggest a convergence over time. To the extent that the response in liver can be compared to muscle there is evidence that the response in conditional KD is time-dependent. The other major issue is that the Hatem-Vaquero is whole body cKD. Whereas Kang (and Trefts) was tissue specific.

Thank you for your comment. On page 13 ILK is mentioned under the integrin section because it is one of the main proteins involved in the integrin pathway, which directly interacts with integrins; on page 22 the topic is focused only on ILK, rather than integrins, then the section involves more details about this protein.

Heading on page 22. Not sure what is meant by "non-pathological context." The studies describe HFD mice and Type 2 diabetes. By "basal-fed" do you mean standard chow?

That is correct, basal-fed correspond to standard chow-fed diet.

p. 23, l. 4 - "ILK is an important kinase". I suggest rephrasing. There is a kinase domain and kinase activity in a test tube. It is contentious as to whether there is kinase activity in cells or whole animal. I believe the consensus is that the ILK kinase domain is a binding site for parvin and some other proteins. ILK is an important protein but probably not for kinase activity.

Thank you, it is indeed an imprecision, we have changed with "ILK is an important effector".

There are two issues that may be relevant to the discussion that the authors should consider touching on.

PINCH is shown in the graph. The initial view going back several decades was that the IPP would affect glucose transport by PINCH-NICK inhibition of insulin signaling. I agree with the case made in the Review. It is more likely to be GTPase and actin-cytoskeleton related. However, I suggest the authors leave the PINCH option in play.

The integrin receptor field is divided (or doesn't really know) whether the more important activator of integrin signaling is by ligand binding or physical force applied to the extracellular domain of the integrin receptor. The latter could be relevant in a condition like muscle contraction and would be an interesting point to acknowledge.

David Wasserman

Many thanks, these are very good and relevant points; however, our review already discusses several topics, thus we think it would be better to keep it concise and leave these two issues for another potential future review.

Dear Dr Mackenzie,

Re: JP-TR-2022-283039R1 "Involvement of the extracellular matrix and integrin signalling proteins in skeletal muscle glucose uptake" by Richard WA Mackenzie, Fulvia Draicchio, Volker Behrends, Neale Anthony Tillin, Nicholas Hurren, and Lykke Sylow

I am pleased to tell you that your Topical Review article has been accepted for publication in The Journal of Physiology, subject to any modifications to the text that may be required by the Journal Office to conform to House rules.

NEW POLICY: In order to improve the transparency of its peer review process The Journal of Physiology publishes online as supporting information the peer review history of all articles accepted for publication. Readers will have access to decision letters, including all Editors' comments and referee reports, for each version of the manuscript and any author responses to peer review comments. Referees can decide whether or not they wish to be named on the peer review history document.

The last Word version of the paper submitted will be used by the Production Editors to prepare your proof. When this is ready you will receive an email containing a link to Wiley's Online Proofing System. The proof should be checked and corrected as quickly as possible.

All queries at proof stage should be sent to tjp@wiley.com.

The accepted version of the manuscript will be published online, prior to copy editing in the Accepted Articles section.

Are you on Twitter? Once your paper is online, why not share your achievement with your followers. Please tag The Journal (@jphysiol) in any tweets and we will share your accepted paper with our 22,000+ followers!

Yours sincerely,

Ian D. Forsythe
Deputy Editor-in-Chief
The Journal of Physiology
<https://jp.msubmit.net>
<http://jp.physoc.org>
The Physiological Society
Hodgkin Huxley House
30 Farringdon Lane
London, EC1R 3AW
UK
<http://www.physoc.org>
<http://journals.physoc.org>

*** IMPORTANT NOTICE ABOUT OPEN ACCESS ***

To assist authors whose funding agencies mandate public access to published research findings sooner than 12 months after publication The Journal of Physiology allows authors to pay an open access (OA) fee to have their papers made freely available immediately on publication.

You will receive an email from Wiley with details on how to register or log-in to Wiley Authors Services where you will be able to place an OnlineOpen order.

You can check if your funder or institution has a Wiley Open Access Account here <https://authorservices.wiley.com/author-resources/Journal-Authors/licensing-and-open-access/open-access/author-compliance-tool.html>

Your article will be made Open Access upon publication, or as soon as payment is received.

If you wish to put your paper on an OA website such as PMC or UKPMC or your institutional repository within 12 months of publication you must pay the open access fee, which covers the cost of publication.

OnlineOpen articles are deposited in PubMed Central (PMC) and PMC mirror sites. Authors of OnlineOpen articles are permitted to post the final, published PDF of their article on a website, institutional repository, or other free public server, immediately on publication.

Note to NIH-funded authors: The Journal of Physiology is published on PMC 12 months after publication, NIH-funded authors DO NOT NEED to pay to publish and DO NOT NEED to post their accepted papers on PMC.

EDITOR COMMENTS

Reviewing Editor:

Thank you for addressing the comments raised in review and congratulations on an insightful manuscript.

Senior Editor:

Thank you for an interesting review.

1st Confidential Review

21-Jul-2022